# Rapid Diagnosis of *Pseudomonas aeruginosa* in Wounds with Point-Of-Care Fluorescence Imaing

**DOI:** 10.3390/diagnostics11020280

**Published:** 2021-02-11

**Authors:** Rose Raizman, William Little, Allie Clinton Smith

**Affiliations:** 1Department of Professional Practice, Scarborough Health Network, Lawrence S. Bloomberg Faculty of Nursing, University of Toronto, Toronto, ON M1E 4B9, Canada; 2Department of Honors Studies, Texas Tech University, Lubbock, TX 79409, USA; william_little@outlook.com (W.L.); allie.c.smith@ttu.edu (A.C.S.)

**Keywords:** bacteria, fluorescence imaging device, *Pseudomonas aeruginosa*, chronic wounds

## Abstract

*Pseudomonas aeruginosa* (PA) is a common bacterial pathogen in chronic wounds known for its propensity to form biofilms and evade conventional treatment methods. Early detection of PA in wounds is critical to the mitigation of more severe wound outcomes. Point-of-care bacterial fluorescence imaging illuminates wounds with safe, violet light, triggering the production of cyan fluorescence from PA. A prospective single blind clinical study was conducted to determine the positive predictive value (PPV) of cyan fluorescence for the detection of PA in wounds. Bacterial fluorescence using the MolecuLight *i:X* imaging device revealed cyan fluorescence signal in 28 chronic wounds, including venous leg ulcers, surgical wounds, diabetic foot ulcers and other wound types. To correlate the cyan signal to the presence of PA, wound regions positive for cyan fluorescence were sampled via curettage. A semi-quantitative culture analysis of curettage samples confirmed the presence of PA in 26/28 wounds, resulting in a PPV of 92.9%. The bacterial load of PA from cyan-positive regions ranged from light to heavy. Less than 20% of wounds that were positive for PA exhibited the classic symptoms of PA infection. These findings suggest that cyan detected on fluorescence images can be used to reliably predict bacteria, specifically PA at the point-of-care.

## 1. Introduction

It is well established that bacterial proliferation in wounds contributes to infection and delayed wound healing [1,2,3,4]. The chronic wound microenvironment is ideal for bioburden and usually contains multiple bacterial species [5]. Among the most common bacterial species observed in chronic wounds is the gram-negative opportunistic pathogen *Pseudomonas aeruginosa* (PA) [6,7]. This non-fermenting aerobic rod bacterial species is a common pathogen in nosocomial infections, particularly in chronic and burn wounds [8,9,10]. Immunocompromised patients and those with comorbidities such as vascular diseases and diabetes are particularly susceptible to developing PA infections [11]. In a multicenter retrospective analysis of 970 venous leg ulcers, PA was detected in one third of patients [12], while another study reports the prevalence of PA in more than half of the chronic leg ulcers evaluated [9]. The presence of PA in wounds is associated with more severe wound outcomes. Wounds containing PA are often larger in size and experience prolonged duration compared to wounds that do not contain PA [5,7,13,14,15]. The presence of PA in burn wounds results in more rapid deterioration [16] and higher rates of sepsis leading to death [17]. Similarly, in leg ulcers, the presence of PA was associated with larger size, delayed healing [13] and a higher rate of skin graft failure [4]. The pathogenicity of PA is mediated, in part, by its capacity to produce a variety of virulence factors that mitigate the impact of environmental stressors and xenobiotic agents [18]. PA is notorious for its intrinsic resistance to many antibiotics and ability to form biofilm matrices that evade conventional antibiotics [19,20,21,22]. Due to the limited treatment options available and the potential for this pathogen to develop antibiotic resistance, it has been identified by the World Health Organization as a critical research priority for the development of novel therapies [23]. 

Currently, point-of-care identification of wounds infected with PA is typically through evaluation of clinical signs and symptoms. Wounds grossly colonized or infected with PA may present with a malodorous, greenish crust, a greenish tinge on removed dressings or may emit a characteristic sweet odor [24,25]. These characteristics are attributed in part to the production of pyocyanin, a blue–green pigment that causes oxidative stress in the host [26]. However, in many wounds with PA infection, these symptoms are not observed [27]. Moreover, high bacterial loads may not mount these symptoms, thus delaying detection and onset of treatment. In most cases, PA infection in wounds is confirmed by culture-dependent microbiological analysis. These results often arrive 24–48 h after sampling, delaying application of necessary treatments to reduce the PA burden. New approaches to rapidly detect PA infection in wounds are needed to mitigate the challenges associated with treating PA infection.

One way to detect PA more rapidly in wounds is by taking advantage of its intrinsic fluorescence properties. Studies have shown that *Pseudomonas spp.* produce a unique cyan fluorescent signature under violet light illumination [28,29,30], attributed to the endogenous production of pyoverdine, a fluorescent siderophore [31,32]. Pyoverdine is unique to the Pseudomonad family and acts primarily as an iron-scavenging molecule in PA [33]. It is one of several virulence factors produced by PA that is also involved in regulating the production of several secreted toxins [34]. Pyoverdine plays an essential role in establishing infection and biofilm formation [35], and its production is directly linked to PA proliferation and virulence against mammalian and invertebrate hosts [36,37,38,39]. Additional studies have shown that the accumulation of pyoverdine correlates with virulence of PA, attributed in part to resistance to multiple antimicrobials [40]. As PA is the most common Pseudomonad detected in wounds [5,27], the fluorescent properties of pyoverdine may be exploited in the clinical context to serve as a key biomarker of PA infection. 

### Evidence for Cyan Fluorescence Detected from Pseudomonas aeruginosa

Preclinical and clinical studies have consistently reported detection of cyan fluorescence correlating with the presence of PA when imaged with violet light (MolecuLight *i:X,* Toronto, ON Canada). Compelling in vitro data have reported the detection of cyan fluorescence from PA (ATCC 9027) as early as 24 h after being incubated in blood agar plate media [41] (Figure 1a). Similarly, in mouse wounds inoculated with PA, strong cyan fluorescence was detected as early as 1 day after inoculation, and this fluorescence signature persisted up to 8 days after inoculation [42] (Figure 1b). In an egg-based infection model, real-time fluorescence imaging detected cyan fluorescence indicative of PA, and this was used to assess the efficacy of antimicrobial treatments against PA [43]. 

In line with preclinical studies, numerous observational studies and clinical trials have detected cyan fluorescence in wounds positive for PA (Figure 1c,d) [27,44,45,46,47,48,49,50,51]. Wound biopsies [27,46], curettage samples [45] or swabs [47,48,49,50,51] were collected for microbiological analysis to confirm the presence and amount of PA detected from cyan positive regions. In multiple instances, cyan fluorescence indicative of PA was detected in wounds otherwise lacking the typical signs of PA infection [27,44,46]. Venous leg ulcers and surgical sites were among the most common chronic wounds where cyan fluorescence has been detected (Figure 1c,d), consistent with the known high prevalence of PA in leg ulcers [5]. In one study examining fluorescence detection from chronic wounds in an outpatient plastic surgery center, the presence of PA was reported in all wounds where cyan fluorescence was detected, resulting in a sensitivity and specificity of 100% [49]. These results contrast with those of Pjipe et al. who used fluorescence imaging in a population of burn wounds: nine of which were deemed positive for cyan fluorescence [47]. In this study, cyan fluorescence resulted in a sensitivity of 100% for detection of PA, yet positive predictive value (PPV) was only 44% [47]. This discrepancy was attributed to inaccurate sampling methods and challenges in distinguishing between cyan fluorescence and green fluorescence from tissue, a challenge which may be lessened with experience and training on image interpretation as with any imaging modality [47]. Given the relatively small sample size in both studies, and the variable sampling methods, additional studies are needed to determine the PPV of cyan fluorescence detected under violet light illumination. Therefore, the purpose of the current study was to (1) evaluate the positive predictive value of cyan fluorescence observed on fluorescence images to predict the presence of *Pseudomonas aeruginosa in* chronic wounds and (2) determine the range of bacterial loads detected from cyan-positive regions of the wound. 

## 2. Materials and Methods 

### 2.1. Patient Population

Participants were recruited from the Scarborough Health Network (Centenary site, Toronto, ON, Canada) as part of a non-randomized, single-center evaluation. Ethical approval was granted by the Scarborough and Rouge Hospital Research Ethics Board (SUR-17-011, August 2017). All patients 18 years or older who present with a chronic wound (>4 weeks duration), were able to provide consent and were receiving standard treatment, were eligible to participate. Patients were excluded if they had any known contraindications to routine wound care, had received treatment with an investigational drug within the previous month or were unable to provide consent. All patients provided written consent for participation and medical photography release. 

### 2.2. Fluorescence Imaging 

The fluorescence imaging procedure was performed as part of routine wound assessment. The procedure was explained to all patients before imaging. All wound dressings were removed, and initial cleaning and debridement was performed to remove surface contaminants. Both standard and fluorescence images were acquired for each wound. The imaging was performed as follows: First, a standard image was taken with the imaging device under ambient lighting. The room lights were then turned off or a DarkDrape (MolecuLight Inc., Toronto, ON, Canada) was used to create the darkened environment required to capture a fluorescence image. The 405 nm excitation light emitting diodes (LEDs) were then turned on, the device was placed 8–10 cm away from the wound (guided by the device rangefinder) and a fluorescence image was captured. Specialized optical filters on the device ensure that only relevant fluorescence from tissue and bacteria are captured and the violet excitation light is filtered out. Under violet light illumination, background tissues appear various shades of green, while bacteria at loads >10^4^ CFU/g appear red or cyan [27,52,53]. The study clinician was trained to interpret images and detect the presence of cyan fluorescence using criteria listed in Table 1. 

### 2.3. Sampling and Bacterial Quantification 

To determine if bacteria were present in the regions with cyan fluorescence, a wound sample was collected via curettage sampling [54]. This sub-surface sampling method was important, as studies have demonstrated the ability of fluorescence to detect sub-surface pathogens that swabs may miss [45,53]. Semi-quantitative culture analysis and sensitivity testing was then performed on wound samples, as per standard four-quadrant practice at Shared Hospital Laboratory Inc. (Toronto, ON, Canada). 

### 2.4. Sample Size Calculation and Data Analysis

A priori sample size calculation was performed. With a hypothesized positive predictive value of 0.90, the sample size needed is *n* = 25 to achieve a 10% margin of error at a 90% confidence level. The sample size calculation is based on asymptotic normal approximation, which is given by: (1)$$n\;=\;\frac{\left(\left(\frac{z\alpha }{2}\right)\right)2\;p\left(1-p\right))}{\mathrm{ME}2}$$
where 1 − α is the confidence level, ME is the margin of error, *p* is the planned estimate of positive predictive value, and zα⁄2 is the 100(1 − α/2) percentile of the standard normal distribution.

A curettage scraping obtained from a cyan fluorescence positive area, as indicated by fluorescence imaging, resulting in a microbiology finding confirming the presence of PA (light growth or higher) was considered a true positive (TP). A biopsy or curettage scraping obtained from a cyan fluorescence positive area, as indicated by the imaging, resulting in a microbiology report negative for the presence of PA or with scant growth only was considered a false positive (FP). The positive predictive value—the probability that a region of cyan fluorescence within or around a wound will contain PA—was calculated as follows:(2)$$\mathrm{PPV}\;=\;\frac{\mathrm{TP}}{\left(\mathrm{TP}\;+\;\mathrm{FP}\right)}$$

## 3. Results

A total of 28 patients (Fitzpatrick skin type 1 to 6) with at least one wound emitting cyan on the fluorescence image were recruited into the study (Table 2). Most study participants were male (57.1%), and most wounds were on the lower extremities. Venous leg ulcers (VLUs) were the most common wound type (60.7%). 

### 3.1. Positive Predictive Value and Bacterial Loads of Cyan Fluorescence 

All wounds included in the study fulfilled the criteria for cyan fluorescence outlined in Table 1. Bright white/cyan fluorescence was detected from various wound types that were positive for cyan fluorescence (Figure 2). Under standard light conditions, only one wound portrayed the greenish crust, attributed to pyocyanin, that is commonly associated with presence of PA [26]. Most study wounds demonstrated few “classic” symptoms of PA. Semi-quantitative analysis of curettage samples taken from the region of cyan fluorescence confirmed presence of PA in 26 wounds (true positives); two false positives were detected. This resulted in a positive predictive value (PPV) of 92.9% for cyan fluorescence indicating presence of PA.

Bacterial load of curettage scrapings taken from cyan-positive regions of the wound and analyzed by semiquantitative analysis ranged from scant to heavy growth but were predominantly moderate-to-heavy. Culture analysis confirmed presence of PA in 26/28 wounds. PA was the only Pseudomonad detected in all culture reports. Eighteen (64.3%) wounds had heavy growth of PA, corresponding to cyan-positive regions, while in six (21.4%) wounds, moderate growth of PA corresponding to cyan fluorescence was observed (Figure 3). In two wounds, light growth of PA was observed, while in two wounds semi quantitative analysis revealed scant or no growth of PA from the cyan-positive curettage sample.

### 3.2. Other Microbiological Findings

Consistent with previous studies [5,7,55], PA occurred polymicrobially in 92.3% (24/28) of wounds; in 69% of these wounds, faint red fluorescence indicative of porphyrin-producing bacteria was also observed. The most prevalent species detected in wounds positive for PA include: *Enterococcus faecalis* (34.6% of wounds), *Staphylococcus aureus* (19.2%) and *Escherichia coli* (15.4%). Culture analysis of wound samples revealed antibiotic resistant PA present in three patients, one of which had multi-drug resistant PA. 

### 3.3. Assessment of “Classic” Pseudomonas spp. Symptoms

The presence of the “classic” Pseudomonas *spp.* symptoms (i.e., greenish tinge exudate, sweet smell) in these wounds was surprisingly rare. The study clinician found these symptoms could not be used as a preliminary screening tool to determine which wounds should be imaged for potential study inclusion. Notably, the greenish tinge on the bandage was observed in less than 20% of study wounds, including the example in Figure 2a. A sweet smell was not observed. 

### 3.4. Cyan Signal Monitored after Mechanical and Antimicrobial Treatments

Though not a pre-defined study outcome, the cyan signal from PA after targeted treatments was monitored using fluorescence imaging as part of the patients’ standard of care. In some wounds, once PA presence and location(s) were detected, cleansing and debridement were able to remove the cyan signal, entirely or in part, during the patient’s visit. Other wounds had antimicrobial dressings prescribed, and at the next visit a reduction in cyan signal was typically, but not always, noted. 

The case in Figure 4 provides an example of how detection of cyan fluorescence impacted clincian decision making. In this case, a 78 year old female patient receiving home care was referred to the wound care clinic for treatment of a chronic venous leg ulcer on her left leg. Home care nurses had stopped providing negative pressure wound therapy and antimicrobials to treat the wound, and it had consequently deterioriated. Upon initial examination, no overt signs or symptoms of infection were detected. However, fluorescence images revealed the presence of bright white cyan fluorescence in the periwound region (Figure 4b); this prompted the clinican to perform additional debridement of the wound. After debridement, another standard and fluorescence image was captured, revealing cyan fluorescence and a substantial decrease in bright white/cyan signal previously observed (Figure 4d). The persistence of cyan fluorescence, particularly in the periwound region, prompted the clincian to select a silver cream and dressing and provide a larger dressing to cover the periwound region. 

## 4. Discussion

The need for definitive methods to improve bacterial detection at point-of-care is indisputable, given that >80% of wounds with high bacterial loads are missed by routine assessment [27,56]. The pathogenicity of PA and its propensity for biofilm formation [19,20,21,22] make its rapid detection critical. In this study, PA was detected at point-of-care in regions of cyan fluorescence with a PPV of 93%. The PPV reported here is similar to findings from Hurley et al. [49], but with a larger sample size and across a greater variety of wound types in a statistically powered trial. In contrast, the PPV we report here is double that reported by Pjipe et al. [47]; this is likely due to more targeted sampling to regions of cyan fluorescence and more appropriate subsurface sampling method employed in our study. In addition, the quality of clinical observations and image interpretation may have been higher in our study due to more than 3 years of routine use of the imaging device [45,52,57]. The color red on fluorescence images has been repeatedly shown in clinical trials to have a PPV >95% for detecting high bacterial loads [27,49,53]. Consistent with these findings, the data from this study suggest that cyan on fluorescence images can also be used to reliably predict bacteria, specifically PA. 

In this study, a high variation in the intensity of cyan fluorescence was observed across wounds, in some cases appearing blue–green and in others, appearing a glowing white. The case in Figure 4 highlights how, in an individual wound, the glowing white signal reduces to a pure cyan as debridement or other antimicrobial treatments that presumably lessened the bacterial load are applied. The ability to monitor this change in cyan fluorescence immediately after applying treatment enables real-time insight of the effectiveness of various treatments. Yet, the variation in signal intensity from one wound to another in this study could not entirely be explained by differences in PA load. Indeed, every wound is a unique environment, and it is well established that production of cyan fluorescing pyoverdine is environment specific [28,29,30]. Production of pyoverdine is regulated by a number of environmental factors including iron concentration, biofilm formation and bacterial cell aggregation [58,59]. Furthermore, the wound microbiome can change over time [60,61,62], and PA is opportunistic and tends to colonize wounds where competing bacteria have recently been eliminated, e.g., through antibiotics [62]. In these instances, the fluorescence images would likely show a switch from red fluorescing bacteria to cyan. 

It was not surprising that PA was the only Pseudomonad detected in this study; larger clinical trials have shown that PA is the overwhelmingly predominant Pseudomonad in chronic wounds [5,27]. Pseudomonads, in particular PA, are known producers of red fluorescing porphyrins [63]; when grown in vitro in conditions that favor porphyrin production, red fluorescence is readily detected from PA [64], evident in Figure 1a. Yet clinically we observed that the red fluorescence appears to be dwarfed by the strong cyan pyoverdine signal. Pyoverdine production is enhanced under hostile conditions (e.g., low iron) [58,65,66]; for example, an increase in pyoverdine production has been observed in burn wounds [18]. Wound debridement and application of antimicrobial agents may further exacerbate this hostile environment, promoting pyoverdine synthesis. Wounds in our study were chronic in nature and therefore some had been treated previously with antimicrobials, making them more likely to have this hostile environment triggering high pyoverdine synthesis. Whether PA would ever express solely porphyrins, not pyoverdines, in a wound environment has, to our knowledge, not been studied. If this were to occur then red fluorescence would presumably be used to guide bacterial removal treatments, as it does for other pathogens [27,45,67]. Given the association between pyoverdine production and PA virulence [36,37,38,39], detection of wounds in which PA is expressing this virulence factor has large clinical relevance. 

### Limitations

There are several limitations of this study which should be noted. This trial was designed to test the positive predictive value of cyan on fluorescence images, thus it does not provide information on negative predictive value, sensitivity or specificity of the images. These have been assessed by other studies, which demonstrate high sensitivity and specificity of the images for detecting high bacterial loads [27,44,46,49]. Future work that examines the presence of PA from wound regions positive or negative for cyan fluorescence may help to clarify the specificity of cyan for PA. In addition, the semi-quantitative culture based microbiological confirmation used has inherent limitations, as this method can underestimate bacterial loads and each semi-quantitative category is associated with a wide range of CFU/g counts. For example, light growth has been shown to range from 10^3^ to 10^6^ CFU/g in wound samples [68]. Underestimating bacterial loads may have resulted in a slightly lower reported PPV in this study, as scant growth of PA was detected in one clearly cyan-positive wound and was considered a false positive in this analysis. Additional studies utilizing gold standard quantitative or molecular culture-based methods to analyze wound biopsies may help to clarify the bacterial loads detected from regions of cyan fluorescence. The fluorescence imaging procedure also has inherent limitations, namely the need for darkness during imaging and a limited depth of excitation (~1.5 mm) for detection of subsurface bacteria. However, as PA tends to be a surface or immediately sub-surface pathogen [69,70], this poses less of a limitation for the detection of PA than it may be for other pathogens. 

Pyoverdine, the virulence factor and source of cyan signal, plays an essential role in establishing infection and biofilm formation [35]. However, the percentage of study wounds in which cyan fluorescence was indicative of regions of PA in biofilm could not be determined due to lack of available methods for this confirmation. Fluorescence signals can come from bacteria in biofilm or in planktonic form [27,42,64]. PA is notorious for its propensity to form biofilm [20,21,22] and, given the known frequency of biofilm in PA-positive wounds, it is likely that bacteria in biofilm were a contributing factor to the cyan signal in many of these wounds. 

## 5. Clinical Implications and Conclusions

PA infection is a common occurrence in chronic wounds [71] and is more fatal than other bacterial infections if left undetected or inappropriately treated [72]. In this study, we show that fluorescence imaging can rapidly detect PA in chronic wounds. With a 93% PPV, point-of-care fluorescence imaging provides an opportunity to accurately locate and detect presence of PA early, before infection potentially spreads. Clinical guidelines advise against wound culture unless infection is suspected [73], therefore wound clinicians commonly rely on the classic *Pseudomonas spp.* symptoms to aid early detection of PA [24]. In practice however, these symptoms may not mount in infected wounds. Indeed, less than 20% of wounds in this study that were positive for PA exhibited these symptoms. Furthermore, most wounds with heavy growth of PA on the culture reports did not exhibit these symptoms. As such, clinicians may be missing the majority of wounds containing PA; clinician reliance on these symptoms for detection of PA is clearly inadequate. 

Treatment of PA continues to be a challenge given the lack of information available at the point-of-care on the efficacy of treatments used to eradicate PA, propensity for antibiotic overuse to manage skin and soft tissue infections [74] and the tendency of PA to rapidly develop resistance to multiple classes of antibiotics [75]. Here we show how point-of-care fluorescence imaging alongside assessment of clinical signs and symptoms of infection, can help to overcome these challenges by reliably enhancing detection of PA in wounds and providing immediate information to wound care clinicians on the efficacy of treatments targeted to eradicate PA.

## Figures and Tables

**Figure 1 diagnostics-11-00280-f001:**
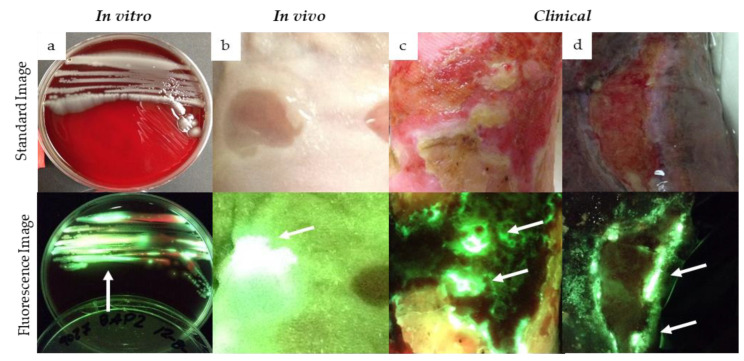
Detection of cyan fluorescence in preclinical and clinical studies using fluorescence imaging. Standard (top panel) and fluorescence (bottom panel) images were captured of preclinical and clinical cases of cyan fluorescence from *Pseudomonas aeruginosa* (PA). Cyan fluorescence was detected 24 h after inoculating blood agar plates with PA (**a**), and 24 h after NCr mouse wounds were inoculated with PA (**b**). Cyan fluorescence has also been observed from chronic wounds positive for PA (**c**,**d**), confirmed by microbiological analysis (clinicaltrials.gov (accessed on 1 February 2021): NCT03540004). White arrows denote regions of cyan fluorescence.

**Figure 2 diagnostics-11-00280-f002:**
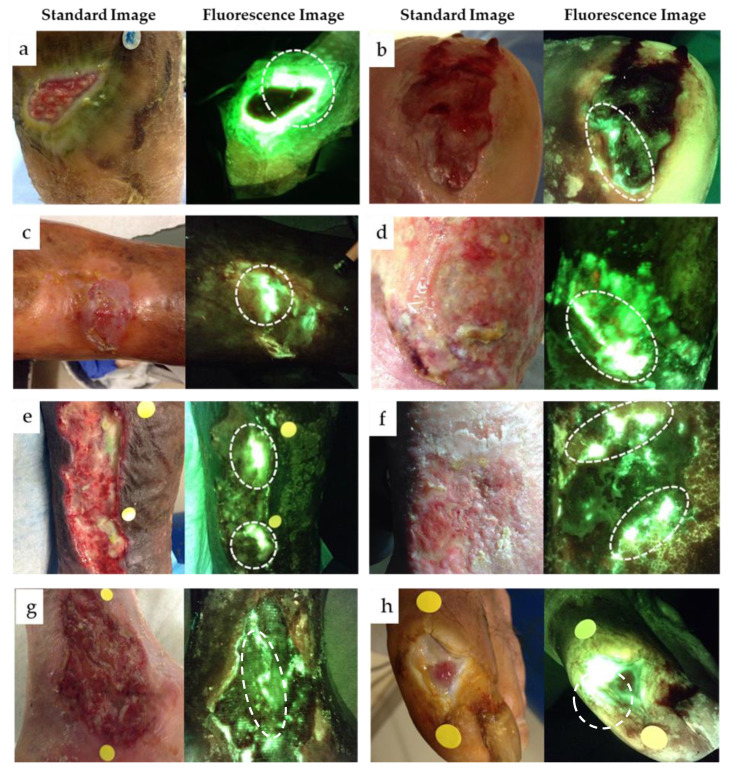
Example standard and fluorescence images of eight patients. Wound types included DFUs (**a**,**b**,**h**), VLUs (**c**,**d**,**f**,**g**) or other (**e**). Among the examples listed, only one wound (**a**) displayed the classic greenish tinge on bandages associated with presence of PA. Each wound had discrete locations of cyan fluorescence. A curettage sample was collected from the region(s) denoted by a dashed white circle in each wound image. Bacterial species and load were identified using semi-quantitative culture analysis. In all images shown here, regions of cyan fluorescence corresponded with presence of PA. Tissue appears green on fluorescence images.

**Figure 3 diagnostics-11-00280-f003:**
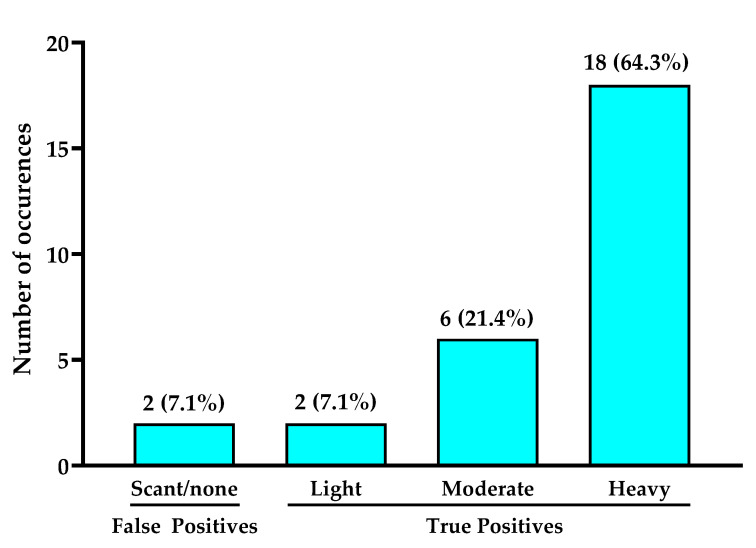
Total bacteria load quantified via semi-quantitative culture analysis from sampled regions of cyan fluorescence. Numbers on top of bars indicate number of study wounds in each category. Wounds where light to heavy bacterial loads were identified were considered true positives. In one wound, cyan fluorescence was observed but microbiological analysis did not detect bacterial load, thus it was considered a false positive.

**Figure 4 diagnostics-11-00280-f004:**
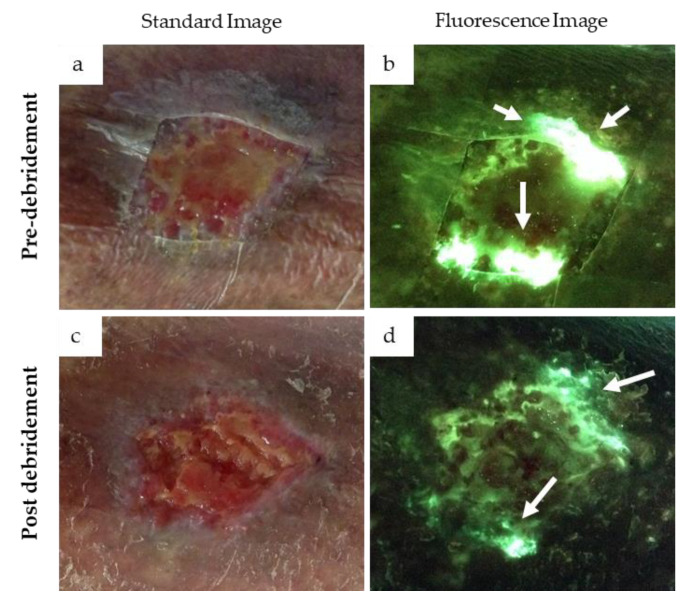
Presence of cyan fluorescence in a 78-year-old female patient with a venous leg ulcer. Standard images (**a**,**c**) and fluorescence images (**b**,**d**) were captured at the point-of-care before and after debridement. Bright white/cyan fluorescence signal was observed before debridement (**b**). Cyan fluorescence signal intensity was significantly reduced after debridement (**d**) but was still clearly present (white arrows).

**Table 1 diagnostics-11-00280-t001:** Questions used to aid in identification of cyan fluorescence in images.

Question	Example Image
**Does the fluorescent signature have a glowing white center with a blue/green border?** Dashed circle outlines regions of glowing cyan/white in the wound.	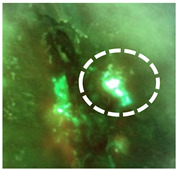
**Do any sharp edges match biological landmarks observed in the standard image?** The bright cyan fluorescence observed (white arrows) does not correspond to any specific landmark or tissue structure on the standard image (yellow arrow).	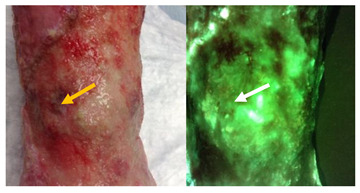
**How does it compare to the color of the surrounding skin?** The surrounding skin tissue will appear a dull green (yellow arrow) compared to the bright white/cyan fluorescence indicative of bacterial burden (white arrows).	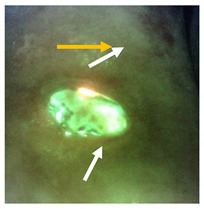

**Table 2 diagnostics-11-00280-t002:** Patient demographics.

Patient Demographics	No. (% of Total)
Sex (male)	16 (57.1)
**Wound Type**	
VLU	17 (60.7%)
DFU	4 (14.3%)
Surgical site	2 (7.1%)
Pressure ulcer	1 (3.6%)
Lymphedema	1 (3.6%)
Rheumatoid wound	1 (3.6%)
Other	2 (7.1%)

## Data Availability

Data available on request from the authors.

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
