# Peer review of "Rapid Diagnosis of Pseudomonas aeruginosa in Wounds with Point-Of-Care Fluorescence Imaing"

_diagnostics, 2021, doi:10.3390/diagnostics11020280_

Round 1

Reviewer 1 Report

Interesting article about the early detection of  Pseudomonas aeruginosa  in wounds, ample introduction, good study methodology, numerous figures that support your results. Congratulations for all these positive aspects. However the fallowing modifications need to be done:

Please specifiy in the last paragraph of the introduction the purpose of the study. Please refer in the discussion to the importance of your findings in certain categories of patients much affected by wounds such as diabetic patientes or cancer patients. Please reffer to the increase resitance of pseudomonas to antibiotics that is caused by enviromental factors and the importance of early detection of the presence of PA in wounds. You can refer to the fallowing articles:

Venter, A.C.; RoÅŸca, E.; Daina, L.G.; MuÅ£iu, G.; Pirte, A.N.; Rahotă, D. Phyllodes tumor: Diagnostic imaging and histopathology findings. Rom. J. Morphol. Embryol. 2015, 56, 1397–1402.

Saramasan C, Identification, Communication And Management Of Risks Relating To Drinking Water Pollution In Bihor County, Environmental Engineering and Management Journal, November/December 2008, Vol.7, No.6, 769-774

Author Response

Thank you for taking the time to review this manuscript. We appreciate your thoughtful suggestions to expand on some key concepts in this work.

We have revised the text in the last paragraph of the introduction to more clearly indicate the purpose of the study as follows:

“Therefore, the purpose of the current study was to: (1) evaluate the positive predictive value of cyan fluorescence observed on fluorescence images to predict the presence of Pseudomonas aeruginosa in chronic wounds and (2) determine the range of bacterial loads detected from cyan-positive regions of the wound.”

In response to your suggestions to expand on the importance of our findings in the discussion, we have added clinical implications to the conclusion with some references to support this point:

Clinical Implications & Conclusions

PA infection is a common occurrence in chronic wounds [72], and is more fatal than other bacterial infections if left undetected or inappropriately treated [73]. In this study, we show that fluorescence imaging can rapidly detect PA in chronic wounds. With a 90% PPV, point-of-care fluorescence imaging provides an opportunity to accurately locate and detect presence of PA early, before infection potentially spreads. Clinical guidelines advise against wound culture unless infection is suspected [74], therefore wound clinicians commonly rely on the classic Pseudomonas spp. symptoms to aid early detection of PA [25]. In practice however, these symptoms may not mount in infected wounds. Indeed, less than 20% of wounds in this study that were positive for PA exhibited these symptoms. Furthermore, most wounds with heavy growth of PA on the culture reports did not exhibit these symptoms. As such, clinicians may be missing the majority of wounds containing PA; clinician reliance on these symptoms for detection of PA is clearly inadequate.

Treatment of PA continues to be a challenge given the lack of information available at the point-of-care on the efficacy of treatments used to eradicate PA, propensity for antibiotic overuse to manage skin and soft tissue infections [75], and the tendency of PA to rapidly develop resistance to multiple classes of antibiotics [76].  Here we show how point-of-care fluorescence imaging alongside assessment of clinical signs and symptoms of infection, can help to overcome these challenges by reliably enhancing detection of PA in wounds and providing immediate information to wound care clinicians on the efficacy of treatments targeted to eradicate PA.

Reviewer 2 Report

The work entitled “Rapid Diagnosis of Pseudomonas aeruginosa in Wounds with Point-of-Care Fluorescence Imaging” by Raizman et al describes a new methodology to successfully and easily identifying the presence of P. aeruginosa in chronic wounds.

The work is well funded with a very complete description of the subject and the goal of this work. The novelty is also very clear.

The presentation of the data is well put together, with many of this Reviewer initial questions being answer as we continue reading the information. The only one that is still not completely clear is in the specificity of the cyan fluorescence to P. aeruginosa. Do the authors have any idea how to improve its specificity, so when other microbial cells are presence and may “cover” PA they can still detect it? They say the color changes a bit in the presence of other microorganisms, but we have seen that on many occasions, particularly in the presence of biofilms, some bacteria are more exposed than others. It would be great, if they could provide a more in-depth analysis into the future and try to come up with solutions for the many limitations of this study.

The presentation of the list of limitations at the end of the manuscript was a very smart move, but they should as well try and come up with solutions for those.

Still, generally this is a very well done and sound work, with a very important need being addressed.  I recommend this manuscript publication after some aspects highlighted earlier are addressed.

Author Response

Thank you for raising these excellent points. This study was designed to evaluate positive predictive value and thus we targeted wound sampling to regions of cyan fluorescence; as such, we were unable to evaluate specificity of the cyan fluorescence to P. aeruginosa. We have instead, provided recommendations on future studies that can address this as follows:
“Future work that examines the presence of PA from wound regions positive or negative for cyan fluorescence may help to clarify the specificity of cyan for PA.”
In addition, we have provided suggested solutions to address the limitations associated with semi-quantitative culture based microbiological analysis as follows:
“Additional studies utilizing gold standard quantitative or molecular culture-based methods to analyze wound biopsies may help to clarify the bacterial loads detected from regions of cyan fluorescence.”
The inherent limitations of the imaging technology (i.e. excitation depth and inability to distinguish bacteria in biofilm or in planktonic form), are beyond the scope of our clinical expertise and as such, we are unable to comment on potential solutions.
